# Comfort in a cross-sector care delivery model to address birth inequities: Learnings from San Francisco's Pregnancy Village

Osamuedeme J. Odiase[1]*, April J. Bell[2], Alison M. El Ayadi[1,3], KaSelah Crockett[4,5], Malini A. Nijagal[1], Patience A. Afulani[1,3]

1 Department of Obstetrics, Gynecology, and Reproductive Sciences, University of California, San Francisco, California, United States of America, 2 Department of Family and Community Medicine, University of California, San Francisco, California, United States of America, 3 Department of Epidemiology and Biostatistics, University of California, San Francisco, California, United States of America, 4 Compass & Keys, Oakland, California, United States of America, 5 Pop-Up Village, Oakland, California, United States of America

* osamuedeme.odiase@ucsf.edu

## Abstract

### Introduction

Comfort is a key factor in physical and mental health, influencing overall well-being. Though once seen as peripheral to the patient care experience, it is now recognized as a critical outcome. For Black and other historically minoritized birthing individuals—who face racism, disproportionly higher morbidities, and unequal care—comfort is key to a safe, positive pregnancy experience. Innovative, community-driven models designed to improve comfort are therefore needed. San Francisco's Pregnancy Village (PV) is a novel, cross-sector care delivery model providing a one-stop shop for clinical, city government, and wraparound services in a dignified and uplifting environment for Black and other minoritized pregnant individuals and their families. This study aims to examine comfort at PV and its key predictors.

### Methods

We surveyed 114 participants (57 pregnant/postpartum individuals and 57 family members) between July 10, 2021 and June 30, 2022. Comfort was measured using a 3-item scale capturing the affective dimension of comfort, with scores standardized to 0–100 scale. Additionally, two individual items assessed situational aspects of comfort: (1) discomfort being seen at PV and (2) feeling out of place at PV. We performed univariate, bivariate, and multivariate analyses.

### Results

The mean comfort score was 96.2/100 ($SD$ = 11.4). Pregnant and postpartum participants, as well as those with limited social support, felt significantly less comfortable

**Data availability statement:** The datasets generated and/or analyzed during the current study are not publicly available due to ethical and privacy restrictions. Although the data have been de-identified, they contain sensitive information from study participants and the combination of variables could potentially allow re-identification. Data may be made available to qualified researchers upon reasonable request and subject to approval by the University of California, San Francisco Institutional Review Board. Requests for access to the data should be directed to the corresponding author or to the University of California, San Francisco Institutional Review Board at irb@ucsf.edu, where requests will be reviewed to ensure that the proposed use is consistent with the ethical approvals and participant consent under which the data were collected.

**Funding:** PAA and MAN received awards from the California Preterm Birth Initiative (A133134; https://pretermbirthca.ucsf.edu/), the California Health Care Foundation (A139605; https://www.chcf.org/), and the Robert Wood Johnson Foundation (80229; https://www.rwjf.org/). The funders were not involved in the study design, data collection, analysis, interpretation, or manuscript preparation.

**Competing interests:** The authors have declared that no competing interests exist.

**Abbreviations:** CALM, Comfort Always Matters Framework; PV, Pregnancy Village; GCQ, General Comfort Questionnaire.

with the idea of being seen by friends at PV compared to family members and those with strong social support, respectively. Participants with some higher education and those reporting occasional everyday discrimination felt significantly less out of place at PV than those with a high school diploma or no discrimination experiences.

## Conclusions

The Pregnancy Village model fostered generally high levels of comfort among Black and other minoritized pregnant individuals and their families in San Francisco, California. However, lower comfort levels among pregnant and postpartum individuals, those with lower educational attainment, and individuals lacking social support underscore the need for greater investment in co-led community-institutional, culturally responsive, and trauma-informed care approaches to foster comfort, particularly for those who face the severest inequities.

---

## Introduction

Comfort is a determinant of physical and mental health, significantly impacting overall well-being [1]. The Comfort Always Matters (CALM) Framework defines comfort as a dynamic, transient state that encompasses relief from physical and emotional distress, accompanied by a growing sense of safety, positivity, and resilience [2]. This state is sustained by feeling valued and adequately supported, and by having agency in care decisions [2]. While comfort has traditionally been considered auxiliary to quality of care models, it has recently emerged as one of the critical patient-reported indicators of experience of care [3]. Black and other historically minoritized women and gender-diverse birthing people (we subsequently use 'individuals' for brevity) in the United States face significant vulnerabilities such as racism and discrimination, disproportionately higher rates of pregnancy complications, and unequal access to high-quality care [4,5]. Ensuring comfort is thus essential to providing a positive and safe pregnancy experience.

Comfort during perinatal care-seeking and receipt is significantly shaped by three interconnected spheres of influence: the individual, interpersonal relationships (e.g., family and friends, and extending to relationships with healthcare personnel), and the clinical built environment [2]. At the individual level, one's sense of comfort is influenced by the use of self-comforting strategies (e.g., seeking signs of safety, positive thinking, and building the capacity to trust), as well as the need for affirming connection through cultural identity, shared community, spiritual practices, or emotionally safe relationships [2]. These forms of connection foster belonging, validate lived experiences, and offer grounding through shared language, rituals, or values, all of which can help individuals regulate emotions and restore a sense of safety and wholeness [6]. Unsurprisingly, family plays a significant role in shaping comfort, providing support rooted in shared culture and mutual understanding [2]. Family involvement and the emotional comfort and security they provide [7] serve as a

protective factor for Black birthing individuals, contributing to reduced stress, lower incidence of postpartum depression, and increased psychological well-being [8].

In the facility setting, patient-provider interactions and the built environment may be equally influential on comfort. The limited existing literature on the comfort levels of Black and other minoritized pregnant individuals highlights their discomfort in care environments through frequent negative care experiences, often shaped by medical and systemic racism. For instance, Black pregnant individuals often report experiencing anti-Black medical gaslighting (e.g., dismissed concerns and credibility being questioned), being treated with less empathy, and receiving differential treatment, where they may be "red flagged" based on their personal history, such as prior contact with the criminal justice system [9–11]. The clinical built environment plays a key role in shaping comfort [1]; yet greater emphasis has historically been placed on designing systems and processes to maximize clinical function and efficiency, while the physical environment has been largely overlooked in consideration of safety, comfort, and well-being for all individuals [12]. This failure falls hardest on those at the margins, particularly Black birthing individuals, who often experience emotional unsafety in clinical spaces [13,14]. This has consequently spurred a growing movement toward community-based perinatal care as a more inclusive and affirming alternative [15,16]. These compounding structural failures underscore the urgent need to design care environments that are safe, accessible, supportive, and comfortable for every person who enters them [17].

Addressing these individual, interpersonal, and structural inequities requires novel, community-driven models designed to foster comfort in care environments to improve the care experience and outcomes for marginalized communities. In the current paper, we evaluate comfort and key predictors of comfort among participants who engaged in the "SF Family & Pregnancy Pop-Up Village" (subsequently referred to as "Pregnancy Village" (PV) for brevity), a cross-sector collaboration providing comprehensive wellness services in a supportive, comforting, and uplifting monthly one-stop shop environment for Black-identifying pregnant individuals in San Francisco. Further information about the development and implementation of PV can be found elsewhere [18,19]. The analysis focuses on the formative phase of PV spanning the initial nine months from July 2021 to June 2022.

## Materials and methods

### Setting

The Pregnancy Village is situated in the Bayview, a historically marginalized neighborhood in San Francisco with approximately 35,000 residents who face significant health inequities. Most birthing individuals are enrolled in Medicaid (61%), and 93% are from racial or ethnic minority groups [20]. The neighborhood experiences significantly lower rates of timely prenatal care, as well as higher rates of preterm births and low birth weight [21]. To promote access and community engagement, PV held events near major public transit routes and community-based organizations [22]. The space was designed with vibrant visual elements, shaded areas, varied seating, and attractive ground treatments to create a welcoming, community-centered environment that reflects PV's core values [22].

### Intervention

The Pregnancy Village is a collaborative, cross-sector care delivery model designed to address perinatal inequities by providing a one-stop shop for comprehensive wellness services in a celebratory, uplifting, and dignified environment for Black pregnant individuals and their families. Held monthly, these events bring together partners from the city government, healthcare, and community-based organizations. The services offered encompass traditional healthcare support—such as medical consultations and Medicaid enrollment—and holistic wellness practices, including food demonstrations, acupuncture, massage, dance classes, and sharing circles [19]. Grounded in anti-racism and person-centered care principles, the model fosters sustainable community-institution partnerships and integrates a real-time feedback mechanism to ensure the model remains responsive to evolving community needs [19]. Further details regarding the implementation of PV can be found elsewhere [19].

## Study design

The data for this analysis are part of a larger community-engaged, mixed-methods evaluation involving quantitative and qualitative data triangulation. The evaluation aimed to assess: 1) the feasibility and fidelity of PV; 2) accessibility and acceptability of PV, and factors influencing sustained participation from service providers; and 3) the preliminary impact of PV, including perceptions of comfort, person-centeredness, and trust [22,23]. This paper reports on the quantitative component of the evaluation, specifically measuring participants' perceived comfort at PV and examining its key predictors.

## Sampling and participants

We employed a convenience sampling strategy to recruit pregnant and postpartum individuals and family members from the first nine monthly PV events (July 10, 2021 through June 30, 2022). Although PV was specifically designed to address perinatal care inequities experienced by Black pregnant and postpartum individuals, the event organizers acknowledged its broader relevance for other marginalized populations facing systemic barriers to care. As such, recruitment efforts focused on engaging Black and other minoritized pregnant/postpartum individuals and families who accessed PV services. Eligibility criteria included: 1) being at least 15 years old for pregnant or postpartum individuals, or 18 years old for family members; 2) participation in at least one PV event; and 3) ability to communicate in English or Spanish. Our recruitment goal was 120 participants in total, with a target of enrolling approximately 10–15 individuals per event based on feasibility.

## Recruitment and data collection

Eligible participants were identified through the PV registration, where study team members explained the study, invited participation, and obtained verbal informed consent. Participants could complete the survey onsite using a tablet or later on their own device through a QR code provided at the event. To minimize potential bias and ensure participants' privacy, surveys were administered in a quiet area of PV. Participants were compensated with a $20 gift card for their participation. Individuals who attended more than one PV event were allowed to complete the survey multiple times, and multiple entries were accounted for in the analysis (see below). Of 104 eligible participants enrolled, 89 completed the survey (response rate 86%). Fifteen participants completed the survey more than once, resulting in a total of 116 survey responses.

## Measures

**Dependent variable: Comfort.** Comfort was operationalized as encompassing both the affective dimension—including pleasant mood, emotional uplift, and perceived safety—and situational aspects, such as feeling out of place or discomfort being seen. Comfort at PV was measured using a three-item scale adapted from the shortened 12-item General Comfort Questionnaire (GCQ) [24], supplemented by two additional items assessing situational aspects of comfort: 'feeling out of place at PV', derived from the shortened GCQ, and 'discomfort being seen at PV', developed by study team members (Individual items are provided in S1 Table). The three-item scale captured affective dimensions of comfort, such as pleasant mood, emotional uplift, and perceived safety, demonstrating good internal consistency in the full sample (Cronbach's α = 0.77). In contrast, the two standalone items assessed situational aspects of comfort related to perceptions of social belonging. Including both measures allowed for a more comprehensive assessment of participants' comfort within the PV environment. However, the latter two items were examined individually rather than combined into a single scale, given that their inter-item reliability was insufficient to support a composite measure. All items had a four-point frequency response option [i.e., 0-("No, not at all"),1-("A little"), 2-("Somewhat"), 3- ("Yes, definitely")]. Negatively worded items (e.g., 'discomfort being seen at PV' and 'feeling out of place at PV') were reverse-coded. Missing data

(2.1%) were imputed using the mean of other items in the measure. The scores were summed and subsequently standardized to range from 0 to 100.

## Covariates

Participants self-reported a range of sociodemographic characteristics, including age, race and ethnicity, gender identity, educational attainment, employment status, primary language, English language proficiency, housing status, residence, relationship status, social support, medical insurance status, food insecurity [25], and receipt of public assistance (see Table 1). Obstetric characteristics were also collected, including pregnancy status (pregnant/postpartum or family member of pregnant/postpartum individual), parity, history of prior preterm birth, pregnancy loss history (e.g., miscarriage, induced abortion, or stillbirth), and prenatal care attendance during the current or most recent pregnancy (for those who were pregnant or postpartum). Additionally, the survey included items assessing past experiences of discrimination, adapted from the Everyday Discrimination Scale and Discrimination in Medical Settings Scale [26,27].

## Analyses

The final analytic sample consisted of 114 responses from 89 unique individuals, after excluding four individuals deemed ineligible on a review of their demographic data. Descriptive statistics were performed to summarize participants' socio-demographic and obstetric characteristics and their reported mean comfort level. Before conducting bivariate analyses, housing status categories were recoded (see Table 3). However, other variables were retained in their original categorical form to preserve nuance in sociodemographic and obstetric characteristics, as collapsing categories would obscure mean-ingful differences within the sample. To maintain statistical power and include as many participants as possible in the mul-tivariate analyses, we recoded missing responses to questions about everyday discrimination and discrimination during prenatal care as "Sometimes." This choice reflects an assumption of moderate exposure to discrimination among those with missing data and helps avoid listwise deletion, which could disproportionately remove individuals facing structural barriers to completing the survey [28]. We estimated linear mixed-effects models for both bivariate and multivariate analy-ses to address clustering due to participants completing the survey multiple times. This method allowed us to account for variability within and between participants, reducing potential bias from repeated measures. Bivariate analyses examined associations between each of the three outcome measures (the overall comfort score and the two individual comfort items ('discomfort being seen at PV' and 'feeling out of place at PV')), and predictor variables including sociodemographic char-acteristics, obstetric factors, and experiences of racism and discrimination. Predictors with p-values <0.05 in the bivariate analyses were included in the multivariate models. The final model selection was informed by checking for collinearity and overall model fit. To assess the robustness of the results, sensitivity analyses were conducted by excluding missing or duplicate responses (i.e., subsequent survey responses from the same individual). All analyses were performed in STATA (version 14) [29], with statistical significance set at $p < 0.05$.

## Ethics approval

All procedures performed in studies involving human participants were in accordance with the ethical standards of the institutional and/or national research committee and with the 1964 Helsinki Declaration and its later amendments or comparable ethical standards. Ethical approval was granted by the Institutional Review Board (IRB) of the University of California, San Francisco (#20–32393). The inclusion of participants under 18 years of age was approved by the IRB through a waiver of parental permission and a waiver of assent. The IRB granted these waivers after determining that the study procedures would not adversely affect the rights or welfare of minor participants. Because the study was conducted in a community setting and aimed to reduce participant burden, researchers obtained verbal informed consent from all participants after explaining the study and before collecting any data. Each participant was given an information sheet

**Table 1. Univariate distribution of predictor variables.**

| Participant Characteristics | Total (*N*=89) | |
| --- | --- | --- |
| | N | % |
| *Age* | | |
| 15-24 | 13 | 14.6 |
| 25-34 | 28 | 31.5 |
| 35-44 | 18 | 20.2 |
| 45 and older | 19 | 21.4 |
| Unknown | 11 | 12.4 |
| *Gender* | | |
| Female | 77 | 86.5 |
| Male | 6 | 6.7 |
| Other/unknown/prefer not to answer | 6 | 6.7 |
| *Race/ethnicity* | | |
| Black | 32 | 36.0 |
| Latine/Hispanic | 34 | 38.2 |
| Multiracial | 17 | 19.1 |
| Other/unknown/prefer not to answer | 6 | 6.7 |
| *Language* | | |
| English | 52 | 58.4 |
| Spanish | 29 | 32.6 |
| Unknown/other/prefer not to answer | 8 | 9.0 |
| *English proficiency* | | |
| Very well or well | 63 | 70.8 |
| With difficulty | 16 | 18.0 |
| Unknown/prefer not to answer | 10 | 11.2 |
| *Pregnancy status* | | |
| Currently/recently pregnant | 47 | 52.8 |
| Family | 42 | 47.2 |
| *Parity* | | |
| No births | 17 | 19.1 |
| 1–2 births | 39 | 43.8 |
| 3 or more births | 25 | 28.1 |
| Unknown/not applicable/ prefer not to answer | 8 | 9.0 |
| *Prenatal care* | | |
| No prenatal care during current/recent pregnancy | 3 | 3.4 |
| Received prenatal care during current/recent pregnancy | 39 | 43.8 |
| Unknown/not applicable/prefer not to answer | 47 | 52.8 |
| *History of preterm birth* | | |
| No preterm births | 59 | 66.3 |
| At least 1 preterm birth | 21 | 23.6 |
| Unknown/prefer not to answer/not applicable | 9 | 10.1 |
| *History of pregnancy loss* | | |
| No prior pregnancy loss | 50 | 56.2 |
| Pregnancy loss | 26 | 29.2 |
| Unknown/prefer not to answer/not applicable | 13 | 14.6 |
| *Education* | | |
| Less than a high school degree | 17 | 19.1 |

*(Continued)*

**Table 1.** (Continued)

| Participant Characteristics | Total (*N*=89) | |
|---|---|---|
| | **N** | **%** |
| High school graduate, GED or equivalent | 23 | 25.8 |
| Some college, junior college or vocational school | 22 | 24.7 |
| College graduate or higher | 19 | 21.4 |
| Unknown/prefer not to answer | 8 | 9.0 |
| *Employment* | | |
| No | 56 | 62.9 |
| Yes, Full-time | 14 | 15.7 |
| Yes, Part-time | 10 | 11.2 |
| Unknown/prefer not to answer | 9 | 10.1 |
| *Income assistance* | | |
| No | 38 | 42.7 |
| Yes | 44 | 49.4 |
| Unknown/prefer not to answer | 7 | 7.9 |
| *Residence* | | |
| Bayview-Hunter's Point | 31 | 34.8 |
| Other San Francisco | 38 | 42.7 |
| East Bay | 10 | 11.2 |
| Other/unknown/prefer not to answer | 10 | 11.2 |
| *Housing status* | | |
| Homeless Shelter | 10 | 11.2 |
| Living with someone for free | 5 | 5.6 |
| No living place | 1 | 1.1 |
| Owns home or apartment | 8 | 9.0 |
| Public Housing | 15 | 16.9 |
| Renting home or apartment | 39 | 43.8 |
| Transitional or supportive housing | 2 | 2.3 |
| Other/unknown/prefer not to answer | 9 | 10.1 |
| *Social support* | | |
| Not at all | 10 | 11.2 |
| A little | 9 | 10.1 |
| Somewhat | 20 | 22.5 |
| Yes, definitely | 48 | 53.9 |
| Unknown/prefer not to answer | 2 | 2.3 |
| *Medical insurance status* | | |
| Private insurance | 15 | 16.9 |
| Public insurance | 58 | 65.2 |
| No insurance | 4 | 4.5 |
| Unknown/prefer not to answer | 12 | 13.5 |
| *Food insecurity (Worried food would run out)* | | |
| Often true | 12 | 13.5 |
| Sometimes true | 22 | 24.7 |
| Never true | 40 | 44.9 |
| Unknown/prefer not to answer | 15 | 16.9 |
| *Food insecurity (Insufficient funds)* | | |

*(Continued)*

**Table 1.** (Continued)

| Participant Characteristics | Total (*N*=89) | |
|---|---|---|
| | N | % |
| Often true | 11 | 12.4 |
| Sometimes true | 27 | 30.3 |
| Never true | 36 | 40.5 |
| Unknown/prefer not to answer | 15 | 16.9 |
| *Relationship status* | | |
| Partnered, living together | 31 | 34.8 |
| Partnered, not living together | 15 | 16.9 |
| Single | 33 | 37.1 |
| Other/unknown/prefer not to answer | 10 | 11.2 |
| *Everyday discrimination experience* | | |
| Never | 17 | 19.1 |
| Rarely | 19 | 21.4 |
| Sometimes | 34 | 38.2 |
| Often | 17 | 19.1 |
| Unknown/prefer not to answer | 2 | 2.3 |
| *Prenatal care discrimination experience* | | |
| Never | 35 | 39.3 |
| Rarely | 17 | 19.1 |
| Sometimes | 23 | 25.8 |
| Often | 8 | 9.0 |
| Unknown/prefer not to answer | 6 | 6.7 |

outlining the study in detail, and trained study personnel recorded verbal consent in paper screening and enrollment logs, with documentation confirming that consent was obtained. The use of verbal informed consent was approved by the IRB.

## Results

### Participant characteristics

Sociodemographic and obstetric characteristics of the 89 unique participants are provided in Table 1 (See S2 Table for univariate distributions by subgroup). Fifty-three per cent of participants were either currently pregnant or recently pregnant (within the preceding 12 months). Approximately half were aged 25–34 years (32%) or 35–44 years (20%). Over one-third identified as Black or African American (36%), and 38% identified as Latine. Forty-six per cent had attained education beyond high school, whereas 19% had not completed high school. A majority were unemployed (63%), and nearly half (49%) reported receiving income assistance. Public health insurance coverage (e.g., Medi-Cal, Medicaid) was reported by 65% of participants. Thirty-seven per cent were single, and 35% resided with a romantic partner. A history of preterm birth was reported by 24% of participants, and 29% had experienced at least one prior pregnancy loss.

### Comfort at PV

Table 2 displays the standardized mean comfort scores from the three-item scale assessing the affective dimension of comfort. The mean comfort score for the full sample was 96.2 (*SD* = 11.4). Pregnant and postpartum participants had a mean score of 95.2 (*SD* = 12.9), while family members had a slightly higher mean of 97.1 (*SD* = 9.7), *p* = 0.3894. By racial and ethnic group, the mean score among Black participants was 96.5 (*SD* = 10.6) compared with 95.9 (*SD* = 12.0) for

**Table 2. Distribution of standardized Comfort Scale score and two items of comfort, N = 114.**

| Measure | Summary statistics | Combined sample | Pregnant and postpartum | Family member | Black participants | Participants from all other racial/ethnic groups |
|---|---|---|---|---|---|---|
| **Comfort scale** | N | 114 | 57 | 57 | 47 | 67 |
| | *Mean* | 96.2 | 95.2 | 97.1 | 96.5 | 95.9 |
| | *SD* | 11.4 | 12.9 | 9.7 | 10.6 | 12.0 |
| | *Minimum Score* | 44.4 | 44.4 | 44.4 | 44.0 | 44.0 |
| | *Maximum Score* | 100.0 | 100.0 | 100.0 | 100.0 | 100.0 |
| **Discomfort being seen at PV** | *N* | 114 | 57 | 57 | 47 | 67 |
| | *Mean* | 2.6 | 2.4 | 2.8 | 2.6 | 2.6 |
| | *SD* | 1.0 | 1.2 | 0.7 | 1.0 | 1.0 |
| | *Minimum Score* | 0.0 | 0.0 | 0.0 | 0.0 | 0.0 |
| | *Maximum Score* | 3.0 | 3.0 | 3.0 | 3.0 | 3.0 |
| **Feeling out of place at PV** | *N* | 114 | 57 | 57 | 47 | 67 |
| | *Mean* | 2.3 | 2.1 | 2.5 | 2.5 | 2.2 |
| | *SD* | 1.1 | 1.2 | 1.0 | 1.1 | 1.2 |
| | *Minimum Score* | 0.0 | 0.0 | 0.0 | 0.0 | 0.0 |
| | *Maximum Score* | 3.0 | 3.0 | 3.0 | 3.0 | 3.0 |

Abbreviations: SD: standard deviation.

participants identifying with other racial and ethnic backgrounds, *p = 0.813*. In addition, the table displays the means (SD) for the two items assessing the situational aspects of comfort. For the item 'discomfort being seen at PV,' the full sample had a mean of 2.6 (*SD* = 1.0) on a 3-point scale. Pregnant and postpartum participants had a lower mean (2.4 [*SD* = 1.2]) than family members (2.8 [*SD* = 0.7]). For the item 'feeling out of place at PV,' the overall mean was 2.3 (*SD* = 1.1), with pregnant and postpartum participants again having slightly lower scores (2.1 [*SD* = 1.2]) than family members (2.5 [*SD* = 1.0]).

### Factors associated with comfort

**Comfort score.** Table 3 shows the bivariate and multivariate analyses of the comfort score. In bivariate analyses, participants without medical insurance, those who did not disclose their food insecurity status, and those who reported occasional discrimination during their usual prenatal care encounters scored on average 14.8, 8.2, and 6.9 points lower, respectively, than those with public insurance, no food insecurity, and no experiences of prenatal care discrimination. Participants aged 45 years and older scored 7.9 points higher, on average, than those aged 15–24 years. In the final multivariate model (Table 4), participants aged 45 years and older scored, on average, 10.7 points higher than those aged 15–24 (95% CI: 1.8, 9.6).

**Situational indicators of comfort. Discomfort being seen at PV:** Table 4 shows the bivariate and multivariate analyses of the discomfort being seen at PV item. In bivariate analyses, pregnant and postpartum participants, as well as those reporting only "somewhat" having social support, scored on average 0.4 and 0.6 points lower, respectively, than family members and those with definite social support ("Yes, definitely"). Conversely, participants aged 45 years and

**Table 3. Bivariate and multivariate mixed-effects linear regression of predictor variables on the Comfort score, N = 114.**

| Predictor variables | Cross Tabs | | Bivariate mixed-effects | | | | Multivariate mixed-effects | | | |
|---|---|---|---|---|---|---|---|---|---|---|
| | *Mean* | *SD* | *β* | *[95% CI]* | | *p-value* | *β* | *[95% CI]* | | *p-value* |
| *Age* | | | | | | | | | | |
| 15–24 [Reference Group] | 91.0 | 19.7 | 0.0 | | | | 0.0 | | | |
| 25 −34 | 94.6 | 14.0 | 3.6 | −3.6 | 10.8 | 0.327 | 0.8 | −7.7 | 9.3 | 0.856 |
| 35 - 44 | 97.8 | 7.2 | 6.8 | −0.7 | 14.2 | 0.075 | 5.7 | −3.6 | 14.9 | 0.230 |
| 45 and older | 98.9 | 3.4 | 7.9 | 0.6 | 15.1 | 0.033 | 10.7 | 1.8 | 19.6 | 0.019 |
| Unknown | 95.6 | 11.7 | 4.5 | −3.7 | 12.8 | 0.282 | 5.2 | −4.8 | 15.3 | 0.309 |
| *Gender* | | | | | | | | | | |
| Female [Reference Group] | 96.4 | 11.4 | 0.0 | | | | | | | |
| Male | 90.7 | 17.8 | −5.6 | −15.0 | 3.7 | 0.235 | | | | |
| Other/unknown/prefer not to answer | 97.2 | 5.1 | 0.8 | −7.3 | 9.0 | 0.841 | | | | |
| *Race/ethnicity* | | | | | | | | | | |
| Black [Reference Group] | 96.5 | 10.6 | 0.0 | | | | | | | |
| Hispanic/Latine | 94.3 | 14.5 | −2.1 | −6.8 | 2.5 | 0.370 | | | | |
| Multiracial | 98.8 | 5.2 | 2.3 | −3.8 | 8.4 | 0.459 | | | | |
| Other/unknown/prefer not to answer | 98.4 | 4.2 | 2.0 | −7.0 | 10.9 | 0.668 | | | | |
| *Language* | | | | | | | | | | |
| English [Reference Group] | 97.3 | 9.2 | 0.0 | | | | | | | |
| Spanish | 93.2 | 15.7 | −4.1 | −8.6 | 0.5 | 0.079 | | | | |
| Other/unknown/prefer not to answer | 98.9 | 3.5 | 1.6 | −5.8 | 9.1 | 0.668 | | | | |
| *English proficiency* | | | | | | | | | | |
| Very well or well [Reference Group] | 96.4 | 11.0 | 0.0 | | | | | | | |
| With difficulty | 97.8 | 7.7 | 1.4 | −4.1 | 6.9 | 0.626 | | | | |
| Unknown/prefer not to answer | 91.7 | 17.8 | −4.7 | −11.6 | 2.1 | 0.173 | | | | |
| *Pregnancy status* | | | | | | | | | | |
| Currently/recently pregnant [Reference Group] | 95.2 | 12.9 | 0.0 | | | | | | | |
| Family | 97.1 | 9.7 | 1.9 | −2.3 | 6.0 | 0.383 | | | | |
| *Parity* | | | | | | | | | | |
| 1–2 births [Reference Group] | 98.0 | 6.2 | 0.0 | | | | | | | |
| None | 93.8 | 16.4 | −4.2 | −9.5 | 1.1 | 0.123 | | | | |
| 3 or more births | 95.6 | 12.2 | −2.4 | −7.6 | 2.8 | 0.373 | | | | |
| Unknown/not applicable/prefer not to answer | 94.4 | 14.1 | −3.6 | −11.2 | 4.1 | 0.361 | | | | |
| *History of preterm birth* | | | | | | | | | | |
| None [Reference Group] | 96.7 | 10.6 | 0.0 | | | | | | | |
| At least 1 preterm birth | 94.7 | 13.3 | −2.1 | −7.3 | 3.2 | 0.445 | | | | |
| Unknown/not applicable/prefer not to answer | 94.9 | 13.5 | −1.8 | −8.9 | 5.4 | 0.624 | | | | |
| *History of pregnancy loss* | | | | | | | | | | |
| None [Reference Group] | 95.6 | 12.9 | 0.0 | | | | | | | |
| Prior pregnancy loss | 98.5 | 6.5 | 2.9 | −2.0 | 7.8 | 0.247 | | | | |
| Unknown/not applicable/prefer not to answer | 94.4 | 11.5 | −1.1 | −7.3 | 5.0 | 0.719 | | | | |
| *Education* | | | | | | | | | | |
| High school graduate, GED, or equivalent [Reference group] | 94.6 | 13.6 | 0.0 | | | | | | | |
| Less than high school degree | 93.1 | 16.3 | −1.5 | −7.7 | 4.7 | 0.634 | | | | |
| Some college, junior college, or vocational school | 99.3 | 3.9 | 4.7 | −0.9 | 10.2 | 0.099 | | | | |

*(Continued)*

| Predictor variables | Cross Tabs | | Bivariate mixed-effects | | | | Multivariate mixed-effects | | | |
|---|---|---|---|---|---|---|---|---|---|---|
| | *Mean* | *SD* | *β* | *[95% CI]* | | *p-value* | *β* | *[95% CI]* | | *p-value* |
| College graduate and professional or graduate school | 97.2 | 8.0 | 2.6 | −3.7 | 8.9 | 0.420 | | | | |
| Unknown/prefer not to answer | 94.9 | 13.5 | 0.3 | −7.4 | 8.0 | 0.935 | | | | |
| *Employment status* | | | | | | | | | | |
| Unemployed [Reference Group] | 96.0 | 12.4 | 0.0 | | | | | | | |
| Full-time | 96.5 | 11.3 | 0.5 | −5.6 | 6.7 | 0.872 | | | | |
| Part-time | 97.2 | 6.9 | 1.2 | −5.7 | 8.1 | 0.734 | | | | |
| Unknown/prefer not to answer | 95.4 | 10.0 | −0.7 | −7.6 | 6.3 | 0.854 | | | | |
| *Income assistance* | | | | | | | | | | |
| No [Reference Group] | 96.2 | 11.1 | 0.0 | | | | | | | |
| Yes | 96.7 | 10.4 | 0.4 | −3.9 | 4.8 | 0.841 | | | | |
| Unknown/prefer not to answer | 92.6 | 18.4 | −3.6 | −11.7 | 4.4 | 0.376 | | | | |
| *Residence* | | | | | | | | | | |
| Bayview-Hunter's Point (SF) [Reference Group] | 97.0 | 10.7 | 0.0 | | | | | | | |
| Other San Francisco | 96.6 | 9.5 | −0.4 | −5.1 | 4.2 | 0.857 | | | | |
| East Bay | 98.0 | 4.5 | 1.0 | −6.5 | 8.4 | 0.799 | | | | |
| Other/unknown/prefer not to answer | 89.8 | 21.4 | −7.2 | −14.4 | 0.0 | 0.051 | | | | |
| *Housing* | | | | | | | | | | |
| Rent home or apartment [Reference group] | 96.1 | 12.0 | 0.0 | | | | | | | |
| Homeless shelter | 100.0 | 0.0 | 3.9 | −3.1 | 10.9 | 0.271 | | | | |
| Owns home or apartment | 95.8 | 8.3 | −0.2 | −8.6 | 8.1 | 0.955 | | | | |
| Public housing | 96.7 | 8.6 | 0.7 | −5.4 | 6.7 | 0.832 | | | | |
| Other (living with someone for free, no living place, transitional housing etc.)/unknown/prefer not to answer | 93.6 | 15.9 | −2.5 | −8.3 | 3.3 | 0.400 | | | | |
| *Social support* | | | 0.0 | | | | | | | |
| Yes, definitely [Reference Group] | 97.7 | 8.0 | | | | | | | | |
| No, not at all | 95.6 | 14.4 | −2.2 | −8.4 | 4.1 | 0.503 | | | | |
| A little | 92.9 | 15.9 | −4.8 | −11.9 | 2.4 | 0.191 | | | | |
| Somewhat | 93.2 | 15.1 | −4.5 | −10.0 | 0.9 | 0.102 | | | | |
| Unknown/prefer not to answer | 100.0 | 0.0 | 2.3 | −10.7 | 15.2 | 0.729 | | | | |
| *Medical insurance status* | | | | | | | | | | |
| Public insurance (e.g., Medi-Cal or Medicaid) [Reference Group] | 97.0 | 10.3 | 0.0 | | | | 0.0 | | | |
| Private or employer provided insurance | 98.8 | 3.6 | 1.7 | −3.9 | 7.3 | 0.545 | 1.9 | −5.2 | 9.1 | 0.597 |
| No insurance | 82.2 | 20.2 | −14.8 | −24.6 | −5.0 | 0.003 | −4.1 | −16.5 | 8.4 | 0.522 |
| Unknown/prefer not to answer | 92.9 | 16.7 | −4.2 | −10.4 | 2.0 | 0.185 | 2.6 | −5.6 | 10.8 | 0.531 |
| *Food insecurity (Worried food would run out)* | | | | | | | | | | |
| Never true [Reference Group] | 97.1 | 10.4 | 0.0 | | | | 0.0 | | | |
| Sometimes true | 97.6 | 7.0 | 0.5 | −4.5 | 5.5 | 0.845 | 0.8 | −8.0 | 9.5 | 0.860 |
| Often true | 98.3 | 4.2 | 1.2 | −5.4 | 7.8 | 0.729 | −2.4 | −14.3 | 9.5 | 0.687 |
| Unknown/prefer not to answer | 88.9 | 19.6 | −8.2 | −14.2 | −2.3 | 0.007 | −3.5 | −13.8 | 6.8 | 0.506 |
| *Food insecurity (Insufficient funds)* | | | | | | | | | | |
| Never true [Reference Group] | 98.0 | 6.2 | 0.0 | | | | 0.0 | | | |
| Sometimes true | 95.0 | 12.9 | −3.0 | −7.9 | 1.9 | 0.226 | −1.2 | −9.9 | 7.5 | 0.791 |
| Often true | 99.1 | 3.1 | 1.1 | −5.6 | 7.9 | 0.740 | −0.4 | −11.9 | 11.2 | 0.949 |

*(Continued)*

**Table 3.** (Continued)

| Predictor variables | Cross Tabs | | Bivariate mixed-effects | | | | Multivariate mixed-effects | | | |
| --- | --- | --- | --- | --- | --- | --- | --- | --- | --- | --- |
| | Mean | SD | β | [95% CI] | | p-value | β | [95% CI] | | p-value |
| Unknown/prefer not to answer | 91.2 | 19.5 | −6.8 | −12.6 | −0.9 | 0.023 | −4.9 | −14.7 | 4.8 | 0.323 |
| *Relationship status* | | | | | | | | | | |
| Married/partnered, living together [Reference Group] | 96.7 | 9.4 | 0.0 | | | | | | | |
| Married/partnered, not living together | 92.4 | 17.6 | −4.3 | −10.9 | 2.2 | 0.193 | | | | |
| Single | 98.1 | 7.7 | 1.4 | −3.4 | 6.1 | 0.570 | | | | |
| Other/unknown/prefer not to answer | 91.7 | 17.8 | −5.0 | −12.3 | 2.2 | 0.173 | | | | |
| *Everyday discrimination* | | | | | | | | | | |
| Never [Reference Group] | 97.2 | 10.1 | 0.0 | | | | | | | |
| Rarely | 96.1 | 13.5 | −1.1 | −7.5 | 5.3 | 0.734 | | | | |
| Sometimes | 94.8 | 12.7 | −2.4 | −8.4 | 3.5 | 0.428 | | | | |
| Often | 98.2 | 4.2 | 1.0 | −6.1 | 8.1 | 0.778 | | | | |
| *Discrimination during prenatal care encounters* | | | | | | | | | | |
| Never [Reference Group] | 98.3 | 7.5 | 0.0 | | | | 0.0 | | | |
| Rarely | 98.5 | 7.1 | 0.2 | −5.3 | 5.7 | 0.947 | 3.1 | −3.7 | 9.8 | 0.374 |
| Sometimes | 91.4 | 16.8 | −6.9 | −11.6 | −2.1 | 0.005 | −2.1 | −8.0 | 3.8 | 0.492 |
| Often | 97.2 | 5.1 | −1.1 | −9.2 | 7.1 | 0.796 | −4.9 | −14.6 | 4.9 | 0.329 |
| Mean dependent var | | | | | | | 90.4 | | | |
| SD dependent var | | | | | | | 13.9 | | | |
| Number of obs | | | | | | | 114 | | | |
| Prob > chi2 | | | | | | | 0.0650 | | | |
| Chi-square | | | | | | | 25.3 | | | |
| Akaike crit. (AIC) | | | | | | | 937.2 | | | |

Abbreviations: SD: standard deviation; 95% CI: 95% Confidence Interval; β: Beta oefficient.

Statistical significance was assessed at α = 0.05.

older scored 0.8 points higher, on average, than those aged 15−24 years. In the final multivariate model, participants who reported only "somewhat" having social support scored 0.5 points lower (95% CI: −1.0, −0.1) than those with definite social support ("Yes, definitely").

**Feeling out of place at PV:** Table 5 shows the bivariate and multivariate analyses of the feeling out of place item. In bivariate analyses, participants working part-time and those who often experienced food insecurity scored, on average, 0.7 and 0.9 points lower, respectively, than those who were unemployed and never experienced food insecurity. Conversely, participants with at least a college degree and those who occasionally experienced everyday discrimination scored 0.7 and 0.6 points higher, respectively, than those with a high school diploma and no discrimination experiences. In the final multivariate model, participants working part-time scored on average 1.0 points lower than those who were unemployed (95% CI: −1.6, −0.4). Those who often experienced food insecurity scored on average 0.7 points lower than those who never did (95% CI: −11.9, 11.2). Participants who reported occasional prenatal discrimination scored 0.6 points higher than those who never experienced such discrimination (95% CI: 0.0,1.1).

## Sensitivity analyses

The sensitivity analysis, which excluded missing and duplicate responses, yielded nearly identical standardized mean comfort scores: 96.4 (SD = 10.6) overall (N = 87), 96.9 (SD = 10.1) for pregnant and postpartum participants (n = 46), and 95.9 (SD = 11.3) for family members (n = 41).

**Table 4. Bivariate and multivariate mixed-effects linear regression of predictor variables on the Discomfort Being Seen at PV item, _N_=114.**

| Predictor variables | Cross Tabs | | Bivariate mixed-effects | | | | Multivariate mixed-effects | | | |
|---|---|---|---|---|---|---|---|---|---|---|
| | _Mean_ | _SD_ | β | _[95% CI]_ | | _p-value_ | β | _[95% CI]_ | | _p-value_ |
| _Age_ | | | | | | | | | | |
| 15–24 [Reference Group] | 2.2 | 1.3 | 0.0 | | | | 0.0 | | | |
| 25 −34 | 2.3 | 1.2 | 0.1 | −0.5 | 0.7 | 0.767 | 0.2 | −0.4 | 0.8 | 0.542 |
| 35 - 44 | 2.6 | 1.0 | 0.4 | −0.2 | 1.0 | 0.203 | 0.4 | −0.2 | 1.0 | 0.155 |
| 45 and older | 3.0 | 0.0 | 0.8 | 0.2 | 1.4 | 0.014 | 0.7 | −0.01 | 1.3 | 0.054 |
| Unknown | 2.5 | 1.1 | 0.3 | −0.4 | 1.0 | 0.395 | 0.04 | −0.7 | 0.8 | 0.914 |
| _Gender_ | | | | | | | | | | |
| Female [Reference Group] | 2.5 | 1.0 | 0.0 | | | | | | | |
| Male | 3.0 | 0.0 | 0.5 | −0.3 | 1.3 | 0.248 | | | | |
| Other/unknown/prefer not to answer | 3.0 | 0.0 | 0.5 | −0.2 | 1.2 | 0.187 | | | | |
| _Race/ethnicity_ | | | | | | | | | | |
| Black [Reference Group] | 2.6 | 1.0 | 0.0 | | | | | | | |
| Hispanic/Latine | 2.6 | 0.9 | 0.0 | −0.4 | 0.4 | 0.910 | | | | |
| Multiracial | 2.3 | 1.3 | −0.3 | −0.8 | 0.3 | 0.329 | | | | |
| Other/unknown/prefer not to answer | 3.0 | 0.0 | 0.4 | −0.4 | 1.2 | 0.304 | | | | |
| _Language_ | | | | | | | | | | |
| English [Reference Group] | 2.6 | 1.0 | 0.0 | | | | | | | |
| Spanish | 2.5 | 1.0 | −0.1 | −0.5 | 0.3 | 0.801 | | | | |
| Other/unknown/prefer not to answer | 2.7 | 0.9 | 0.1 | −0.5 | 0.8 | 0.750 | | | | |
| _English proficiency_ | | | | | | | | | | |
| Very well or well [Reference Group] | 2.5 | 1.1 | 0.0 | | | | | | | |
| With difficulty | 2.7 | 0.8 | 0.1 | −0.4 | 0.6 | 0.678 | | | | |
| Unknown/prefer not to answer | 2.8 | 0.6 | 0.2 | −0.4 | 0.8 | 0.506 | | | | |
| _Pregnancy status_ | | | | | | | | | | |
| Currently/recently pregnant [Reference Group] | 2.4 | 1.2 | 0.0 | | | | 0.0 | | | |
| Family | 2.8 | 0.7 | 0.4 | 0.09 | 0.8 | 0.014 | 0.26 | −0.2 | 0.7 | 0.250 |
| _Parity_ | | | | | | | | | | |
| 1–2 births [Reference Group] | 2.6 | 1.0 | 0.0 | | | | | | | |
| None | 2.2 | 1.3 | −0.4 | −0.9 | 0.0 | 0.061 | | | | |
| 3 or more births | 2.9 | 0.6 | 0.3 | −0.2 | 0.7 | 0.212 | | | | |
| Unknown/not applicable/prefer not to answer | 3.0 | 0.0 | 0.4 | −0.2 | 1.1 | 0.198 | | | | |
| _History of preterm birth_ | | | | | | | | | | |
| None [Reference Group] | 2.5 | 1.1 | 0.0 | | | | | | | |
| At least 1 preterm birth | 2.7 | 0.9 | 0.3 | −0.2 | 0.7 | 0.271 | | | | |
| Unknown/not applicable/prefer not to answer | 3.0 | 0.0 | 0.5 | −0.1 | 1.1 | 0.099 | | | | |
| _History of pregnancy loss_ | | | | | | | | | | |
| None [Reference Group] | 2.5 | 1.0 | 0.0 | | | | | | | |
| Prior pregnancy loss | 2.6 | 1.1 | 0.1 | −0.4 | 0.5 | 0.817 | | | | |
| Unknown/not applicable/prefer not to answer | 2.8 | 0.8 | 0.3 | −0.3 | 0.8 | 0.308 | | | | |
| _Education_ | | | | | | | | | | |
| High school graduate, GED, or equivalent [Reference group] | 2.6 | 0.9 | 0.0 | | | | | | | |
| Less than high school degree | 2.3 | 1.2 | −0.3 | −0.8 | 0.2 | 0.276 | | | | |
| Some college, junior college, or vocational school | 2.5 | 1.1 | −0.1 | −0.6 | 0.4 | 0.678 | | | | |

_(Continued)_

| Predictor variables | Cross Tabs | | Bivariate mixed-effects | | | | Multivariate mixed-effects | | | |
|---|---|---|---|---|---|---|---|---|---|---|
| | *Mean* | *SD* | *β* | *[95% CI]* | | *p-value* | *β* | *[95% CI]* | | *p-value* |
| College graduate and professional or graduate school | 2.9 | 0.7 | 0.2 | −0.3 | 0.8 | 0.438 | | | | |
| Unknown/prefer not to answer | 2.6 | 0.9 | 0.0 | −0.7 | 0.7 | 0.993 | | | | |
| *Employment status* | | | | | | | | | | |
| Unemployed [Reference Group] | 2.5 | 1.0 | 0.0 | | | | | | | |
| Full-time | 2.8 | 0.8 | 0.2 | −0.3 | 0.7 | 0.430 | | | | |
| Part-time | 2.3 | 1.4 | −0.3 | −0.9 | 0.3 | 0.332 | | | | |
| Unknown/prefer not to answer | 3.0 | 0.0 | 0.5 | −0.1 | 1.0 | 0.125 | | | | |
| *Income assistance* | | | | | | | | | | |
| No [Reference Group] | 2.5 | 1.0 | 0.0 | | | | | | | |
| Yes | 2.6 | 0.9 | 0.1 | −0.3 | 0.5 | 0.615 | | | | |
| Unknown/prefer not to answer | 2.6 | 1.0 | 0.0 | −0.7 | 0.7 | 0.965 | | | | |
| *Residence* | | | | | | | | | | |
| Bayview-Hunter's Point (SF) [Reference Group] | 2.7 | 0.8 | 0.0 | | | | | | | |
| Other San Francisco | 2.4 | 1.1 | −0.3 | −0.7 | 0.1 | 0.115 | | | | |
| East Bay | 3.0 | 0.0 | 0.3 | −0.4 | 0.9 | 0.434 | | | | |
| Other/unknown/prefer not to answer | 2.4 | 1.2 | −0.3 | −0.9 | 0.3 | 0.302 | | | | |
| *Housing* | | | | | | | | | | |
| Rent home or apartment [Reference group] | 2.7 | 0.8 | 0.0 | | | | | | | |
| Homeless shelter | 2.8 | 0.9 | 0.1 | −0.5 | 0.7 | 0.801 | | | | |
| Owns home or apartment | 2.6 | 1.1 | 0.0 | −0.8 | 0.7 | 0.897 | | | | |
| Public housing | 2.5 | 1.2 | −0.2 | −0.7 | 0.3 | 0.451 | | | | |
| Other (living with someone for free, no living place, transitional housing etc.)/unknown/prefer not to answer | 2.3 | 1.2 | −0.4 | −0.9 | 0.1 | 0.164 | | | | |
| *Social support* | | | | | | | | | | |
| Yes, definitely [Reference Group] | 2.7 | 0.9 | 0.0 | | | | 0.0 | | | |
| No, not at all | 2.6 | 1.1 | −0.1 | −0.6 | 0.5 | 0.761 | −0.1 | −0.6 | 0.4 | 0.720 |
| A little | 2.9 | 0.3 | 0.2 | −0.4 | 0.8 | 0.464 | 0.4 | −0.2 | 1.0 | 0.155 |
| Somewhat | 2.1 | 1.3 | −0.6 | −1.1 | −0.1 | 0.012 | −0.5 | −1.0 | −0.1 | 0.018 |
| Unknown/prefer not to answer | 3.0 | 0.0 | 0.3 | −0.8 | 1.4 | 0.570 | 0.6 | −0.5 | 1.7 | 0.308 |
| *Medical insurance status* | | | | | | | | | | |
| Public insurance (e.g., Medi-Cal or Medicaid) [Reference Group] | 2.6 | 1.1 | 0.0 | | | | | | | |
| Private or employer provided insurance | 2.8 | 0.7 | 0.2 | −0.3 | 0.7 | 0.391 | | | | |
| No insurance | 2.6 | 0.9 | 0.0 | −0.8 | 0.9 | 0.927 | | | | |
| Unknown/prefer not to answer | 2.5 | 0.9 | −0.1 | −0.6 | 0.5 | 0.837 | | | | |
| *Food insecurity (Worried food would run out)* | | | | | | | | | | |
| Never true [Reference Group] | 2.7 | 0.9 | 0.0 | | | | | | | |
| Sometimes true | 2.4 | 1.1 | −0.3 | −0.8 | 0.1 | 0.130 | | | | |
| Often true | 2.5 | 1.1 | −0.2 | −0.8 | 0.4 | 0.516 | | | | |
| Unknown/prefer not to answer | 2.5 | 1.0 | −0.3 | −0.8 | 0.3 | 0.330 | | | | |
| *Food insecurity (Insufficient funds)* | | | | | | | | | | |
| Never true [Reference Group] | 2.8 | 0.8 | 0.0 | | | | | | | |
| Sometimes true | 2.3 | 1.2 | −0.4 | −0.8 | 0.0 | 0.056 | | | | |
| Often true | 2.8 | 0.8 | 0.0 | −0.6 | 0.6 | 0.975 | | | | |

*(Continued)*

**Table 4.** (Continued)

| Predictor variables | Cross Tabs | | Bivariate mixed-effects | | | | Multivariate mixed-effects | | |
|---|---|---|---|---|---|---|---|---|---|
| | *Mean* | *SD* | *β* | *[95% CI]* | | *p-value* | *β* | *[95% CI]* | *p-value* |
| Unknown/prefer not to answer | 2.4 | 1.1 | −0.3 | −0.8 | 0.2 | 0.191 | | | |
| *Relationship status* | | | | | | | | | |
| Married/partnered, living together [Reference Group] | 2.8 | 0.7 | 0.0 | | | | | | |
| Married/partnered, not living together | 2.4 | 1.2 | −0.4 | −1.0 | 0.2 | 0.158 | | | |
| Single | 2.6 | 1.1 | −0.2 | −0.6 | 0.2 | 0.314 | | | |
| Other/unknown/prefer not to answer | 2.3 | 1.2 | −0.5 | −1.1 | 0.2 | 0.161 | | | |
| *Everyday discrimination* | | | | | | | | | |
| Never [Reference Group] | 2.5 | 1.1 | 0.0 | | | | | | |
| Rarely | 2.5 | 1.1 | 0.0 | −0.6 | 0.5 | 0.906 | | | |
| Sometimes | 2.7 | 0.9 | 0.2 | −0.3 | 0.7 | 0.525 | | | |
| Often | 2.7 | 0.9 | 0.2 | −0.4 | 0.8 | 0.556 | | | |
| *Discrimination during prenatal care encounters* | | | | | | | | | |
| Never [Reference Group] | 2.6 | 1.0 | 0.0 | | | | | | |
| Rarely | 2.5 | 1.2 | −0.2 | −0.7 | 0.3 | 0.530 | | | |
| Sometimes | 2.6 | 0.8 | 0.0 | −0.4 | 0.4 | 0.940 | | | |
| Often | 2.6 | 1.1 | 0.0 | −0.7 | 0.7 | 0.973 | | | |
| Mean dependent var | | | | | | | 2.6 | | |
| SD dependent var | | | | | | | 1.0 | | |
| Number of obs | | | | | | | 114 | | |
| Prob > chi2 | | | | | | | 0.006 | | |
| Chi-square | | | | | | | 23.0 | | |
| Akaike crit. (AIC) | | | | | | | 320.1 | | |

Abbreviations: SD: standard deviation; 95% CI: 95% Confidence Interval; β: Beta Coefficient.

Statistical significance was assessed at α = 0.05.

## Discussion

We aimed to assess participants' perceived comfort at the Pregnancy Village. We found that overall comfort levels were high. However, pregnant or postpartum individuals and those with limited social support were more likely to feel uncomfortable about being seen by friends at PV compared to family members and individuals with strong social support, respectively. Conversely, participants with some higher education and those reporting occasional everyday discrimination were less likely to feel out of place at PV.

To our knowledge, this is the first study to evaluate participant comfort within a co-led community-institutional perinatal care delivery model, and as such, there are no directly comparable studies. Nonetheless, the high reported levels of comfort are consistent with other findings from the PV evaluation, including outcomes related to acceptability and person-centeredness [22,23]. Different factors may have contributed to the high levels of comfort at PV. First, PV's community-centeredness allowed individuals to connect with others of similar sociocultural backgrounds. This sense of shared experience aligns with CALM's emphasis on cultural connection [2] and is further supported by our acceptability assessment of the PV model [22], and evidence from group prenatal care models, such as CenteringPregnancy, which have demonstrated high acceptability among Black and other minoritized birthing individuals, largely attributed to the strong sense of community and belonging that group participation fosters [30]. Sharing circles at PV may also have fostered social connection and emotional solidarity, enhancing comfort by sharing similar challenges and validating each other's experiences. This is supported by evidence from group prenatal care models, in which sharing experiences

**Table 5. Bivariate and multivariate mixed-effects linear regression of predictor variables on the Feeling Out of Place at PV item, _N_ = 114.**

| Predictor variables | Cross Tabs | | Bivariate mixed-effects | | | | Multivariate mixed-effects | | | |
|---|---|---|---|---|---|---|---|---|---|---|
| | _Mean_ | _SD_ | β | _[95% CI]_ | | _p-value_ | β | _[95% CI]_ | | _P-value_ |
| _Age_ | | | | | | | | | | |
| 15–24 [Reference Group] | 2.2 | 1.3 | 0.0 | | | | | | | |
| 25 −34 | 2.0 | 1.2 | −0.3 | −1.0 | 0.5 | 0.471 | | | | |
| 35 - 44 | 2.5 | 1.0 | 0.2 | −0.5 | 1.0 | 0.509 | | | | |
| 45 and older | 2.6 | 0.9 | 0.4 | −0.4 | 1.1 | 0.314 | | | | |
| Unknown | 2.2 | 1.3 | 0.0 | −0.9 | 0.8 | 0.941 | | | | |
| _Gender_ | | | | | | | | | | |
| Female [Reference Group] | 2.4 | 1.1 | 0.0 | | | | | | | |
| Male | 2.2 | 1.3 | −0.2 | −1.1 | 0.7 | 0.665 | | | | |
| Other/unknown/prefer not to answer | 1.6 | 1.4 | −0.7 | −1.5 | 0.1 | 0.069 | | | | |
| _Race/ethnicity_ | | | | | | | | | | |
| Black [Reference Group] | 2.5 | 1.1 | 0.0 | | | | | | | |
| Hispanic/Latine | 2.3 | 1.1 | −0.2 | −0.6 | 0.3 | 0.442 | | | | |
| Multiracial | 2.2 | 1.3 | −0.3 | −0.9 | 0.3 | 0.330 | | | | |
| Other/unknown/prefer not to answer | 1.7 | 1.3 | −0.8 | −1.6 | 0.1 | 0.096 | | | | |
| _Language_ | | | | | | | | | | |
| English [Reference Group] | 2.4 | 1.1 | 0.0 | | | | | | | |
| Spanish | 2.2 | 1.2 | −0.2 | −0.7 | 0.3 | 0.414 | | | | |
| Other/unknown/prefer not to answer | 2.1 | 1.2 | −0.3 | −1.0 | 0.5 | 0.445 | | | | |
| _English proficiency_ | | | | | | | | | | |
| Very well or well [Reference Group] | 2.3 | 1.2 | 0.0 | | | | | | | |
| With difficulty | 2.4 | 1.1 | 0.0 | −0.5 | 0.6 | 0.873 | | | | |
| Unknown/prefer not to answer | 2.3 | 1.1 | −0.1 | −0.7 | 0.6 | 0.875 | | | | |
| _Pregnancy status_ | | | | | | | | | | |
| Currently/recently pregnant [Reference Group] | 2.1 | 1.2 | 0.0 | | | | | | | |
| Family | 2.5 | 1.0 | 0.4 | −0.04 | 0.8 | 0.078 | | | | |
| _Parity_ | | | | | | | | | | |
| 1–2 births [Reference Group] | 2.3 | 1.2 | 0.0 | | | | | | | |
| None | 2.1 | 1.4 | −0.2 | −0.7 | 0.3 | 0.454 | | | | |
| 3 or more births | 2.5 | 0.9 | 0.3 | −0.3 | 0.8 | 0.334 | | | | |
| Unknown/not applicable/prefer not to answer | 2.4 | 1.1 | 0.1 | −0.6 | 0.9 | 0.757 | | | | |
| _History of preterm birth_ | | | | | | | | | | |
| None [Reference Group] | 2.3 | 1.2 | 0.0 | | | | | | | |
| At least 1 preterm birth | 2.4 | 1.1 | 0.2 | −0.4 | 0.7 | 0.550 | | | | |
| Unknown/not applicable/prefer not to answer | 2.3 | 1.1 | 0.0 | −0.7 | 0.7 | 0.995 | | | | |
| _History of pregnancy loss_ | | | | | | | | | | |
| None [Reference Group] | 2.2 | 1.2 | 0.0 | | | | | | | |
| Prior pregnancy loss | 2.4 | 1.0 | 0.2 | −0.3 | 0.7 | 0.419 | | | | |
| Unknown/not applicable/prefer not to answer | 2.3 | 1.1 | 0.1 | −0.5 | 0.7 | 0.833 | | | | |
| _Education_ | | | | | | | | | | |
| High school graduate, GED, or equivalent [Reference group] | 1.9 | 1.4 | 0.0 | | | | 0.0 | | | |
| Less than high school degree | 2.2 | 1.2 | 0.3 | −0.3 | 0.9 | 0.298 | 0.2 | −0.3 | 0.8 | 0.449 |
| Some college, junior college, or vocational school | 2.6 | 1.0 | 0.7 | 0.2 | 1.2 | 0.012 | 0.5 | 0.0 | 1.0 | 0.054 |

_(Continued)_

| Predictor variables | Cross Tabs | | Bivariate mixed-effects | | | | Multivariate mixed-effects | | | |
|---|---|---|---|---|---|---|---|---|---|---|
| | *Mean* | *SD* | *β* | *[95% CI]* | | *p-value* | *β* | *[95% CI]* | | *P-value* |
| College graduate and professional or graduate school | 2.6 | 0.8 | 0.7 | 0.1 | 1.4 | 0.020 | 0.4 | −0.2 | 1.1 | 0.197 |
| Unknown/prefer not to answer | 2.5 | 1.0 | 0.6 | −0.2 | 1.3 | 0.127 | 0.6 | −0.2 | 1.3 | 0.130 |
| *Employment status* | | | | | | | | | | |
| Unemployed [Reference Group] | 2.4 | 1.1 | 0.0 | | | | 0.0 | | | |
| Full-time | 2.8 | 0.6 | 0.4 | −0.2 | 0.9 | 0.233 | 0.3 | −0.4 | 0.9 | 0.437 |
| Part-time | 1.7 | 1.4 | −0.7 | −1.4 | −0.1 | 0.032 | −1.0 | −1.6 | −0.4 | 0.002 |
| Unknown/prefer not to answer | 1.8 | 1.3 | −0.6 | −1.2 | 0.1 | 0.099 | −0.6 | −1.3 | 0.1 | 0.094 |
| *Income assistance* | | | | | | | | | | |
| No [Reference Group] | 2.1 | 1.3 | 0.0 | | | | | | | |
| Yes | 2.4 | 1.0 | 0.3 | −0.1 | 0.7 | 0.177 | | | | |
| Unknown/prefer not to answer | 2.4 | 1.0 | 0.3 | −0.5 | 1.1 | 0.454 | | | | |
| *Residence* | | | | | | | | | | |
| Bayview-Hunter's Point (SF) [Reference Group] | 2.5 | 1.1 | 0.0 | | | | | | | |
| Other San Francisco | 2.1 | 1.2 | −0.4 | −0.8 | 0.1 | 0.102 | | | | |
| East Bay | 2.1 | 0.9 | 0.2 | −0.6 | 0.9 | 0.645 | | | | |
| Other/unknown/prefer not to answer | 2.5 | 0.9 | 0.0 | −0.7 | 0.8 | 0.917 | | | | |
| *Housing* | | | | | | | | | | |
| Rent home or apartment [Reference group] | 2.3 | 1.2 | 0.0 | | | | | | | |
| Homeless shelter | 2.5 | 1.2 | 0.2 | −0.5 | 0.9 | 0.564 | | | | |
| Owns home or apartment | 2.3 | 1.2 | 0.0 | −0.9 | 0.8 | 0.919 | | | | |
| Public housing | 2.2 | 1.2 | −0.1 | −0.7 | 0.6 | 0.853 | | | | |
| Other (living with someone for free, no living place, transitional housing etc.)/unknown/prefer not to answer | 2.3 | 1.0 | 0.0 | −0.6 | 0.6 | 0.939 | | | | |
| *Social support* | | | | | | | | | | |
| Yes, definitely [Reference Group] | 2.4 | 1.1 | 0.0 | | | | | | | |
| No, not at all | 1.8 | 1.4 | −0.6 | −1.2 | 0.0 | 0.062 | | | | |
| A little | 2.4 | 1.0 | 0.0 | −0.7 | 0.7 | 0.927 | | | | |
| Somewhat | 2.3 | 1.0 | −0.1 | −0.6 | 0.5 | 0.776 | | | | |
| Unknown/prefer not to answer | 2.7 | 0.6 | 0.3 | −1.0 | 1.6 | 0.682 | | | | |
| *Medical insurance status* | | | | | | | | | | |
| Public insurance (e.g., Medi-Cal or Medicaid) [Reference Group] | 2.3 | 1.2 | 0.0 | | | | | | | |
| Private or employer provided insurance | 2.3 | 1.2 | 0.0 | −0.6 | 0.6 | 0.944 | | | | |
| No insurance | 2.2 | 1.1 | −0.1 | −1.1 | 0.9 | 0.850 | | | | |
| Unknown/prefer not to answer | 2.4 | 0.9 | 0.1 | −0.5 | 0.8 | 0.693 | | | | |
| *Food insecurity (Worried food would run out)* | | | | | | | | | | |
| Never true [Reference Group] | 2.5 | 1.0 | 0.0 | | | | | | | |
| Sometimes true | 2.3 | 1.2 | −0.3 | −0.8 | 0.3 | 0.331 | | | | |
| Often true | 1.8 | 1.3 | −0.7 | −1.3 | 0.0 | 0.056 | | | | |
| Unknown/prefer not to answer | 2.1 | 1.1 | −0.4 | −1.0 | 0.2 | 0.214 | | | | |
| *Food insecurity (Insufficient funds)* | | | | | | | | | | |
| Never true [Reference Group] | 2.5 | 1.0 | 0.0 | | | | 0.0 | | | |
| Sometimes true | 2.4 | 1.1 | −0.1 | −0.6 | 0.3 | 0.587 | 0.0 | −0.5 | 0.5 | 0.936 |
| Often true | 1.6 | 1.4 | −0.9 | −1.6 | −0.3 | 0.006 | −0.7 | −1.3 | −0.1 | 0.035 |

*(Continued)*

| Predictor variables | Cross Tabs | | Bivariate mixed-effects | | | | Multivariate mixed-effects | | | |
|---|---|---|---|---|---|---|---|---|---|---|
| | Mean | SD | β | [95% CI] | | p-value | β | [95% CI] | | P-value |
| Unknown/prefer not to answer | 2.0 | 1.2 | −0.5 | −1.1 | 0.0 | 0.066 | −0.3 | −0.9 | 0.3 | 0.273 |
| *Relationship status* | | | | | | | | | | |
| Married/partnered, living together [Reference Group] | 2.3 | 1.1 | 0.0 | | | | | | | |
| Married/partnered, not living together | 1.9 | 1.3 | −0.4 | −1.1 | 0.2 | 0.207 | | | | |
| Single | 2.4 | 1.2 | 0.1 | −0.3 | 0.6 | 0.590 | | | | |
| Other/unknown/prefer not to answer | 2.4 | 0.9 | 0.1 | −0.6 | 0.8 | 0.748 | | | | |
| *Everyday discrimination* | | | | | | | | | | |
| Never [Reference Group] | 1.8 | 1.4 | 0.0 | | | | | | | |
| Rarely | 2.6 | 0.9 | 0.9 | 0.2 | 1.5 | 0.007 | | | | |
| Sometimes | 2.4 | 1.0 | 0.6 | 0.0 | 1.2 | 0.040 | | | | |
| Often | 2.3 | 1.2 | 0.6 | −0.1 | 1.3 | 0.108 | | | | |
| *Discrimination during prenatal care encounters* | | | | | | | | | | |
| Never [Reference Group] | 2.3 | 1.2 | 0.0 | | | | 0.0 | | | |
| Rarely | 2.6 | 0.8 | 0.3 | −0.3 | 0.9 | 0.278 | 0.8 | 0.2 | 1.4 | 0.006 |
| Sometimes | 2.2 | 1.1 | −0.1 | −0.6 | 0.4 | 0.607 | 0.6 | 0.0 | 1.1 | 0.033 |
| Often | 1.8 | 1.5 | −0.6 | −1.4 | 0.3 | 0.174 | 0.5 | −0.2 | 1.1 | 0.138 |
| Mean dependent var | | | | | | | 2.3 | | | |
| SD dependent var | | | | | | | 1.1 | | | |
| Number of obs | | | | | | | 114 | | | |
| Prob > chi2 | | | | | | | 0.000 | | | |
| Chi-square | | | | | | | 38.2 | | | |
| Akaike crit. (AIC) | | | | | | | 351.0 | | | |

Abbreviations: SD: standard deviation; 95% CI: 95% Confidence Interval; β: Beta Coefficient.

provided a unique form of reassurance, as participants found comfort in recognizing that their experiences and challenges were widely shared among peers navigating similar journeys [31]. Additionally, PV's commitment to person-centered care may have contributed to participants feeling welcomed, seen, and valued by care providers [23], highlighting the quality of interpersonal care interactions in positively influencing comfort. Lastly, PV's built environment may have played a key role. The vibrant, colorful tents, varied shade and ground treatments, and culturally affirming music and food likely contributed to a sense of place and belonging, potentially increasing participants' comfort. Moreover, by shifting care delivery from institutional settings—often associated with deep mistrust [32,33]—to community-trusted spaces, PV has physically and symbolically transformed the care landscape.

An unexpected finding was that pregnant and postpartum participants felt less comfortable about being seen by friends at PV than by family members, despite PV's mission to center, support, and celebrate pregnant and postpartum individuals. This discomfort may stem from a fear of being judged for perceived challenges or shortcomings, such as their readiness for parenthood, health, or socioeconomic status [34]. Further, such perceptions may reflect internalized stigmatization and contribute to discomfort seeking help, even from programs and interventions designed to be supportive and affirming. It is important to note that although situational comfort perceptions differed significantly, the absolute difference was marginal, suggesting that PV broadly supports participants' situational comfort. Nevertheless, it is possible that PV is not fully achieving its mission for all participants. These subtle variations thus highlight opportunities to enhance inclusivity and better support vulnerable or marginalized participants who may experience situational discomfort.

Individuals with limited social support also felt less comfortable being seen at PV than those with strong social support. This may be attributed to the challenges of navigating the PV environment alone, without the presence of family or friends who typically offer emotional security, such as guidance and validation [7,8]. Social support plays a critical role in shaping positive pregnancy experiences and outcomes [8,35], and its absence can leave individuals feeling vulnerable, alienated, and stressed, negatively impacting their psychological well-being [8,36]. Further, social support is multidimensional, spanning emotional, informational, instrumental, and appraisal support [37]. While findings on the acceptability of PV suggest that PV generally met informational (e.g., perinatal and nutritional information) and instrumental (e.g., perinatal care resources) support needs [22], PV may not adequately meet the support needs of participants requiring appraisal support (e.g., validation, affirmation) and emotional support (e.g., having someone to confide in).

Individuals with only a high school diploma were more likely to feel out of place at PV than those with some post-secondary education. This could be attributed to inadequate service navigation support at PV. Individuals with lower educational attainment are more likely to have lower health literacy, which can limit their ability to navigate, understand, and act on complex information [38]. Consequently, they may require more intensive support to access care and services effectively. Compounding this issue is the possibility of unconscious provider bias, in which providers may assume that patients with lower educational attainment inherently have limited health literacy, an assumption that can lead to oversimplified explanations and the adoption of more paternalistic care approaches [39]. Without adequate support, these individuals may experience confusion, overwhelm, disempowerment, and discomfort during care encounters [40]. Findings from our evaluation of the accessibility and acceptability of PV support this notion: participants who reported inadequate service navigation perceived PV as less acceptable [22]. In response, we actively sought to expand and improve navigation support within PV. The makeup of providers could also have contributed to a diminished sense of place and belonging. While PV includes a diverse representation of providers, individuals with lower educational attainment may still perceive a lack of representation from providers who share similar socioeconomic experiences, potentially contributing to their reduced comfort.

Notably, those who experienced everyday discrimination felt less out of place than those who never did. This may be attributable to PV's intentional anti-racist approach. Additionally, PV's deliberate framing of care through the lens of comprehensive wellness and empowerment—eschewing the pathologization of individuals' unique circumstances and instead affirming their lived experiences—may mitigate apprehension about being discriminated against for perceived problems. This experience is emblematic of broader discriminatory and inequitable systems [41].

### Strengths and limitations

This study has some limitations. First, the model's consistency fluctuated over time because the evaluation was conducted in a real-world setting, and the model's dynamic co-creation and iterative approach enabled it to adapt to the community's needs. Second, while our sample closely represented the target population, using convenience sampling methods limits the generalizability of our findings. Third, comfort at PV was measured using a brief, three-item adapted scale and two individual items. This may not fully capture all relevant dimensions of comfort in the PV context, underscoring the need for more contextually appropriate comfort measures in future research. Finally, completing surveys onsite may have contributed to social desirability bias. We mitigated this through self-administration, and participants were assured of their confidentiality and anonymity. A key strength of this study is that, to our knowledge, it is the first to assess perceived comfort in such an intervention, highlighting the need for further research in this area.

### Implications

The findings demonstrate the feasibility of creating care environments that are not only accessible but comforting, particularly for those from historically minoritized communities. Black placemaking—rooted in the intersectionality of Blackness, structure, place, and agency, and which involves transforming spaces of occupancy into spaces of cultural affirmation, celebration, and belonging—is essential for fostering comfort and well-being [42–45]. This transformation can be achieved

by integrating culturally grounded practices, such as "Blessingway" ceremonies and birthing affirmations, which help individuals feel seen, valued, and respected [46–48]. Moreover, providing platforms for Black birthing individuals, elders, and doulas to share their lived experiences can serve as a vital means for bolstering comfort [49]. Empowerment initiatives such as birthing rights education, self-advocacy support, and the collaborative development of birth plans [47,50] may also promote comfort. It is also essential that providers reflect the communities they serve socioculturally to make care more comfortable and build trust.

While PV was primarily designed to center Black birthing individuals, it is equally essential to ensure that other minoritized individuals feel comfortable accessing care, namely with the adequate provision of bilingual Spanish-speaking providers and community health workers who possess deep cultural understanding and serve as trusted advocates within their communities. Additionally, lower comfort levels among participants without social support underscore the need for care models to meet not only informational and instrumental support needs but also emotional support needs.

The success of the Pregnancy Village model in fostering high levels of comfort has meaningful implications for public health policy and service delivery reform. Integrating cross-sector partnerships that blend health, social services, and community-led programming at a policy level can shift care from transactional to transformative. This model illustrates the feasibility of reimagining perinatal care through a racial equity and healing-centered lens, which may be particularly beneficial for cities and counties with documented birth inequities. Scaling such interventions will require sustained investment, structural flexibility, and governance models that center community expertise. Embedding funding streams for culturally responsive infrastructure, trusted messengers, and community ownership will be critical for replicating and sustaining this model beyond San Francisco.

## Conclusions

The pilot implementation of the Pregnancy Village model, the SF Family and Pregnancy Pop-Up Village, fostered generally high levels of comfort among Black and other minoritized pregnant individuals and their families in San Francisco, California. These findings underscore the importance of centering Black and other minoritized individuals in reimagining and co-creating what care should *look* and *feel* like, with the goal of transforming the healthcare delivery system. Our findings suggest a critical need to tailor prevalent models of perinatal care delivery to better support pregnant and postpartum individuals, those with lower educational attainment, and those with inadequate social support. While PV has taken meaningful steps to foster a supportive and affirming care environment for Black pregnant individuals, continued investment in community-driven, culturally responsive, and trauma-informed approaches is key to promoting comfort, particularly for those who face the severest inequities.

## Supporting information

**S1 Table. Distribution of comfort scale items.** This table shows the distribution of responses for each item on the comfort scale, as well as the two situational items of comfort.
(XLSX)

**S2 Table. Univariate distribution of predictor variables by subgroup.** This table summarizes the sociodemographic and obstetric characteristics and distribution of predictor variables stratified by subgroup.
(XLSX)

## Acknowledgments

We express our sincere gratitude to the organizations, staff, and volunteers who support PV events, delivering critical care and services to San Francisco's underserved communities. We also wish to acknowledge the invaluable contributions of all PV participants.

## Author contributions

**Conceptualization:** April J. Bell, Alison M. El Ayadi, Malini A. Nijagal, Patience A. Afulani.

**Data curation:** Osamuedeme J. Odiase, April J. Bell, Alison M. El Ayadi, Patience A. Afulani.

**Formal analysis:** Osamuedeme J. Odiase.

**Funding acquisition:** Malini A. Nijagal, Patience A. Afulani.

**Investigation:** Osamuedeme J. Odiase, April J. Bell, Alison M. El Ayadi, Malini A. Nijagal, Patience A. Afulani.

**Methodology:** Osamuedeme J. Odiase, April J. Bell, Alison M. El Ayadi, Malini A. Nijagal, Patience A. Afulani.

**Project administration:** Osamuedeme J. Odiase, April J. Bell, Alison M. El Ayadi, Malini A. Nijagal, Patience A. Afulani.

**Resources:** Malini A. Nijagal, Patience A. Afulani.

**Supervision:** Patience A. Afulani.

**Validation:** Osamuedeme J. Odiase, April J. Bell, Alison M. El Ayadi, Malini A. Nijagal, Patience A. Afulani.

**Writing – original draft:** Osamuedeme J. Odiase.

**Writing – review & editing:** Osamuedeme J. Odiase, April J. Bell, Alison M. El Ayadi, KaSelah Crockett, Malini A. Nijagal, Patience A. Afulani.

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
