## [Decision Letter · Decision Letter 0]

20 Jan 2026

Dear Dr. Odiase,

Thank you for submitting your manuscript to PLOS ONE. After careful consideration, we feel that it has merit but does not fully meet PLOS ONE’s publication criteria as it currently stands. Therefore, we invite you to submit a revised version of the manuscript that addresses the points raised during the review process.

We look forward to receiving your revised manuscript.

Kind regards,

Kwang-Sig Lee

Academic Editor

PLOS One

Journal Requirements:

2. In the ethics statement in the Methods, you have specified that verbal consent was obtained. Please provide additional details regarding how this consent was documented and witnessed, and state whether this was approved by the IRB.

Reviewers' comments:

Reviewer's Responses to Questions

**Comments to the Author**

1. Is the manuscript technically sound, and do the data support the conclusions?

Reviewer #1: Yes

2. Has the statistical analysis been performed appropriately and rigorously?

Reviewer #1: Yes

3. Have the authors made all data underlying the findings in their manuscript fully available?

Reviewer #1: Yes

4. Is the manuscript presented in an intelligible fashion and written in standard English?

Reviewer #1: No

Reviewer #1: Thank you for your submission. The PV program is one of great possibilities, and it was wonderful to see the available evaluation data.

The manuscript has excellent potential with some major revisions. Mainly, the data presentation requires more context to show the reader why the evidence matters.

Below are some specific examples where the data presentation can be improved in the narrative.

Additionally, I strongly suggest that the authors review the tables to assess what data are needed within the manuscript. For example, should the family members be compared to the pregnant/postpartum people in all of the models? If so, why? I appreciate the need to keep a larger sample, but the combination of both groups requires a stronger argument.

Thank you again, and I look forward to seeing the revision.

Page 10 Line 54: describe the specific aspects in the methods

Page 13, Line 122-124: Is it built for the patient safety? Consider a disability justice lens here.

Page 13, Line 124 Are people looking for social interaction in a clinical space? If so, why?

Line 192-3: What privacy measures were offered to the participants on-site to address potential bias?

Line 204: Describe comfort dimensions before introducing the specific items.

Line 209: Describe “hang well”

Line 235-237: Provide a citation to support this assertion

Line 262: change “participants” to “responses” or another word to clarify the differences between number in the sample to the number of times the survey was completed

Line 274: Introduce Table 1 at the beginning of the section so that the reader can follow it with the narrative

Table 1: see my comments above regarding comparison across individuals for all variables.

Line 283: remind the reader of the scale

Table 3: If presenting the p value with the CI, include significance threshold. I suggest recoding most of these variables to summarize the data more succinctly. If the authors disagree, please note within the manuscript the rationale.

Discussion

Lines 353-356: Be clear on the comparison groups.

Lines 358-360: Although this is a novel model, what other comparisons can be made? What about other models of care, like group models? Are there other comfort studies? Please see how more context can be presented to show the value of the model in the larger scope.

Lines 377-378: The difference was significant, yet minor. Does the absolute difference matter? And if so, is there a possibility that the PV is not meeting its mission and changes need to be made?

Line 383: Is it possible that the PV model is not supportive, even with the intentional design? See also lines 392 to 396, lines 402-403 for the same question.

Lines 409-411: Is there a provider bias towards people with lower educational attainment? Is more provider training needed? This is an important consideration for implementation.

.

Reviewer #1: No

You may also use PLOS’s free figure tool, NAAS, to help you prepare publication quality figures: https://journals.plos.org/plosone/s/figures#loc-tools-for-figure-preparation

---

## [Author Response · Author response to Decision Letter 1]

6 Mar 2026

Editor

1)Please ensure that your manuscript meets PLOS ONE's style requirements, including those for file naming.

-We have ensured that the manuscript meets PLOS ONE’s style requirements.

2) In the ethics statement in the Methods, you have specified that verbal consent was obtained. Please provide additional details regarding how this consent was documented and witnessed, and state whether this was approved by the IRB.

- Thank you for this comment. We have revised the ethics statement in the Methods section to provide additional details regarding the verbal consent process.

The revised text reads as follows: “Each participant was given an information sheet outlining the study in detail, and trained study personnel recorded verbal consent in paper screening and enrollment logs, with documentation confirming that consent was obtained. The use of verbal consent was approved by the IRB.”

3) We note that you have indicated that there are restrictions to data sharing for this study. PLOS only allows data to be available upon request if there are legal or ethical restrictions on sharing data publicly. For more information on unacceptable data access restrictions, please see http://journals.plos.org/plosone/s/data-availability#loc-unacceptable-data-access-restrictions.

- The dataset used in this study contains sensitive participant information collected from a vulnerable population. Although the data has been de-identified, the combination of demographic, obstetric, and contextual variables could potentially allow re-identification of participants. Data may be made available to qualified researchers upon reasonable request and subject to approval by the Institutional Review Board at the University of California, San Francisco. Requests for data access may be directed to irb@ucsf.edu, where they will be reviewed to ensure that the proposed use complied with the ethical approvals and participant consent under which the data were collected.

Reviewer #1

Thank you for your submission. The PV program is one of great possibilities, and it was wonderful to see the available evaluation data.

The manuscript has excellent potential with some major revisions. Mainly, the data presentation requires more context to show the reader why the evidence matters.

Below are some specific examples where the data presentation can be improved in the narrative.

Additionally, I strongly suggest that the authors review the tables to assess what data are needed within the manuscript. For example, should the family members be compared to the pregnant/postpartum people in all of the models? If so, why? I appreciate the need to keep a larger sample, but the combination of both groups requires a stronger argument.

Thank you again, and I look forward to seeing the revision.

- We appreciate your engagement with our paper and your thoughtful, constructive feedback.

We agree that comparing family members with pregnant/postpartum individuals in the models requires a strong rationale. By adding family members as a comparator in the models, we were able to achieve the following : 1) assess differences in perceptions and experiences between target participants (pregnant/postpartum) and potentially influential family members; and 2) inform intervention adaptation by highlighting areas where misalignment between the target population (pregnant/postpartum) and family members may limit impact. Further, we believe our approach is justified because it helps preserve statistical power while enabling formal testing for group differences, rather than assuming that family members’ perceptions mirror those of pregnant/postpartum individuals.

Table 1 has been revised to present the sociodemographic and obstetric characteristics of the full sample only. The univariate distribution of predictor variables stratified by subgroup has been relocated to Supplementary Table S2 for readers wishing to examine differences by pregnancy status (pregnant/postpartum vs. family member). Pregnancy status is now presented row-wise in Table 1, in keeping with the format used in Tables 3 and 4. For this variable, the pregnant/postpartum group now serves as the reference group in both the bivariate and multivariate analyses.

2) Page 10 Line 54: describe the specific aspects in the methods

- Thank you for your comment. To clarify, comfort was measured using a 3-item scale designed to capture the affective dimension of comfort. Two additional items that measured situational aspects of comfort captured: 1) discomfort being seen at PV and 2) feeling out of place at PV. We have now clarified this in the methods.

3) Page 13, Line 122-124: Is it built for the patient safety? Consider a disability justice lens here.

Thank you for this comment. While the clinical built environment plays a key role in patients’ experience, we have now emphasized that traditional healthcare design has historically foregrounded systems and processes to maximize clinical function and efficiency. As a result, aspects of the environment that support safety, comfort, and well-being are often overlooked. In relation to your following comment, we have touched upon disability justice principles to expose the lack of emotional safety Black birthing people experience, highlight promising alternatives (community-based models), and call for the urgent need to design care environments that are safe, accessible, supportive, and comfortable for all people.

The revised text reads as follows: “The clinical built environment plays a key role in shaping comfort (1); yet greater emphasis has historically been placed on designing systems and processes to maximize clinical function and efficiency, while the physical environment has been largely overlooked in consideration of safety, comfort, and well-being for all individuals. (12). This failure falls hardest on those at the margins, particularly Black birthing people, who often experience emotional unsafety in clinical spaces (13,14). This has consequently spurred a growing movement toward community-based perinatal care as a more inclusive and affirming alternative (15,16). These compounding structural failures underscore the urgent need to design care environments that are safe, accessible, supportive, and comfortable for every person who enters them (17).”

4) Page 13, Line 124 Are people looking for social interaction in a clinical space? If so, why?

Thank you for raising this important point. We have revised the text to foreground the emotional unsafety Black birthing people experience in clinical spaces, which has, in turn, fueled a growing movement toward community-based perinatal care and, more broadly, underscore the urgent need to design care environment that are safe, accessible, comfortable, and affirming for every birthing person.

5)Line 192-3: What privacy measures were offered to the participants on-site to address potential bias?

Thank you for this comment. To minimize potential bias and ensure participants’ privacy, surveys were administered in a quiet areas of PV.

6)Line 204: Describe comfort dimensions before introducing the specific items.

We appreciate the suggestion. We have briefly defined the comfort dimensions before detailing the scale and respective items.

7) Line 209: Describe “hang well”

Thank you for pointing this out. We have clarified in the manuscript that the two situational aspects of comfort were analyzed separately, as their inter-item reliability was insufficient to support a composite scale.

8) Line 235-237: Provide a citation to support this assertion

Thank you for your comment. We have provided a citation to support this assertion:

Sterne J A C, White I R, Carlin J B, Spratt M, Royston P, Kenward M G et al. Multiple imputation for missing data in epidemiological and clinical research: potential and pitfalls BMJ 2009; 338 :b2393 doi:10.1136/bmj.b2393

9)Line 262: change “participants” to “responses” or another word to clarify the differences between number in the sample to the number of times the survey was completed

We appreciate the suggestion. We have replaced ‘participants’ with ‘responses’.

10) Line 274: Introduce Table 1 at the beginning of the section so that the reader can follow it with the narrative

Thank you for the suggestion. We have introduced Table 1 at the beginning of the section.

11) Table 1: see my comments above regarding comparison across individuals for all variables.

Thank you for raising this point. Table 1 has been revised to present the sociodemographic and obstetric characteristics of the full sample only. The univariate distribution of predictor variables stratified by subgroup has been relocated to Supplementary Table S2 for readers wishing to examine differences by pregnancy status (pregnant/postpartum vs. family member). Pregnancy status is now presented row-wise in Table 1, in keeping with the format used in Tables 3 and 4.

12)Line 283: remind the reader of the scale

We appreciate the suggestion. We have included mention of the comfort scale, which assesses the affective dimension of comfort. The revised text reads as follows:

“Table 2 displays the standardized mean comfort scores from the three-item scale assessing the affective dimension of comfort.”

13)Table 3: If presenting the p value with the CI, include significance threshold. I suggest recoding most of these variables to summarize the data more succinctly. If the authors disagree, please note within the manuscript the rationale.

Thank you for pointing this out. We have included a footnote under Table 3, stating that significance was assessed at α = 0.05.

We carefully considered recoding these variables to provide a more succinct summary. However, we elected to retain the original categories because collapsing them would obscure important nuances in sociodemographic and reproductive/obstetric differences within our sample. Given the study’s focus on understanding differences in experiences across a diverse sample, preserving the full range of responses categories allows for more precise and accurate representation of participants characteristics and how these factors shaped their experience. We have added clarification in the manuscript to explain this rationale:

“However, all other variables were retained in their original categorical form to preserve nuance in sociodemographic and obstetric characteristics, as collapsing categories would obscure meaningful differences within the sample.”

14) Lines 353-356: Be clear on the comparison groups.

Thank you for pointing this out. We have clarified the comparison groups. The revised text reads as follows:

“However, pregnant or postpartum individuals and those with limited social support were more likely to feel uncomfortable about being seen by friends at PV compared to family members and individuals with strong social support, respectively.”

15) Lines 358-360: Although this is a novel model, what other comparisons can be made? What about other models of care, like group models? Are there other comfort studies? Please see how more context can be presented to show the value of the model in the larger scope.

While no other studies, to our knowledge, have measured comfort in similar models, we have situated our findings within the broader group care literature by comparing acceptability, under which comfort can be subsumed. For instance, we theorized that community-centeredness of PV contributed to high comfort, as evidenced by our acceptability findings, and consistent with evidence from group-based models demonstrating high acceptability was rooted, in part, in a strong sense of community.

16) Lines 377-378: The difference was significant, yet minor. Does the absolute difference matter? And if so, is there a possibility that the PV is not meeting its mission and changes need to be made?

We appreciate your thoughtful observation and comments. We do agree that, while the observed difference was statistically significant, the absolute magnitude is nonetheless marginal. Nevertheless, even small differences can be meaningful, particularly for marginalized and vulnerable populations.

Further, we have clarified that, while the findings indicate generally positive situational comfort, PV may not have fully achieved its mission for all participants. These results underscore the need for program improvements aimed at enhancing inclusivity and support.

We have clarified this, as well as contextualized the significance of small differences in the comfort scores and their potential implications for practice.

“It is important to note that although situational comfort perceptions differed significantly, the absolute difference was marginal, suggesting that PV broadly supports participants’ situational comfort. Nevertheless, it is possible that PV is not fully achieving its mission for all participants. These subtle variations thus highlight opportunities to enhance inclusivity and better support vulnerable or marginalized participants who may experience situational discomfort.”

17) Line 383: Is it possible that the PV model is not supportive, even with the intentional design? See also lines 392 to 396, lines 402-403 for the same question.

Thank you for raising this point. In line with your earlier comment, we have acknowledged the possibility that PV may not fully achieve its mission for all participants. Specifically, regarding the social support finding, we have revised the text to note that PV may not adequately meet the support needs of participants requiring appraisal or emotional support.

The revised text reads as follows: “While findings on the acceptability of PV suggest that PV generally met informational (e.g., perinatal and nutritional information) and instrumental (e.g., perinatal care resources) support needs (22), PV may not adequately meet the support needs of participants requiring appraisal support (e.g., validation, affirmation) and emotional support (e.g., having someone to confide in).”

18)Lines 409-411: Is there a provider bias towards people with lower educational attainment? Is more provider training needed? This is an important consideration for implementation.

Thank you for raising this important point. We have now acknowledged this as a potential interpretation of the finding. The revised text reads as follows:

“Compounding this issue is the possibility of unconscious provider bias, where providers may assume that patients with lower educational attainment inherently possess limited health literacy, an assumption that can lead to oversimplified explanations and more paternalistic care approaches (37)”

---

## [Decision Letter · Decision Letter 1]

31 Mar 2026

Comfort in a cross-sector care delivery model to address birth inequities: Learnings from San Francisco’s Pregnancy Village

PONE-D-25-62311R1

Dear Dr. Odiase,

We’re pleased to inform you that your manuscript has been judged scientifically suitable for publication and will be formally accepted for publication once it meets all outstanding technical requirements.

Kind regards,

Kwang-Sig Lee

Academic Editor

PLOS One

Additional Editor Comments (optional):

Reviewers' comments:

Reviewer's Responses to Questions

**Comments to the Author**

Reviewer #1: All comments have been addressed

2. Is the manuscript technically sound, and do the data support the conclusions?

Reviewer #1: Yes

3. Has the statistical analysis been performed appropriately and rigorously?

Reviewer #1: Yes

4. Have the authors made all data underlying the findings in their manuscript fully available?

Reviewer #1: Yes

5. Is the manuscript presented in an intelligible fashion and written in standard English?

Reviewer #1: Yes

Reviewer #1: Thank you for your careful and comprehensive adjustments. Congratulations on an excellent manuscript.

.

Reviewer #1: No

---

## [Editor Report · Acceptance letter]

PONE-D-25-62311R1

PLOS One

Dear Dr. Odiase,

I'm pleased to inform you that your manuscript has been deemed suitable for publication in PLOS One. Congratulations! Your manuscript is now being handed over to our production team.

Kind regards,

on behalf of

Professor Kwang-Sig Lee

Academic Editor

PLOS One